# Modified Montmorillonite Improved Growth Performance of Broilers by Modulating Intestinal Microbiota and Enhancing Intestinal Barriers, Anti-Inflammatory Response, and Antioxidative Capacity

**DOI:** 10.3390/antiox11091799

**Published:** 2022-09-13

**Authors:** Qi Wang, Xiaoli Zhan, Baikui Wang, Fei Wang, Yuanhao Zhou, Shujie Xu, Xiang Li, Li Tang, Qian Jin, Weifen Li, Li Gong, Aikun Fu

**Affiliations:** 1Key Laboratory of Animal Molecular Nutrition of Education of Ministry, National Engineering Laboratory of Biological Feed Safety and Pollution Prevention and Control, Key Laboratory of Animal Feed and Nutrition of Zhejiang Province, Institute of Animal Nutrition and Feed Sciences, College of Animal Sciences, Zhejiang University, Hangzhou 310058, China; 2Zhejiang Fenghong Biological Technology Co., Ltd., Huzhou 313000, China; 3Hainan Institute, Zhejiang University, Yongyou Industry Park, Yazhou Bay Sci-Tech City, Sanya 572000, China; 4School of Life Science and Engineering, Foshan University, Foshan 528225, China

**Keywords:** modified montmorillonite (MMT), broilers, growth performance, gut microbiota, TLRs/MAPK/NF-κB pathway, antioxidative capacity

## Abstract

This study aims to explore the effects of modified montmorillonite (MMT, copper loading) on the growth performance, gut microbiota, intestinal barrier, antioxidative capacity and immune function of broilers. Yellow-feathered broilers were randomly divided into control (CTR), modified montmorillonite (MMT), and antibiotic (ANTI) groups. Results revealed that MMT supplementation increased the BW and ADG and decreased the F/R during the 63-day experiment period. 16S rRNA sequencing showed that MMT modulated the cecal microbiota composition of broilers by increasing the relative abundance of two phyla (Firmicutes and Bacteroidetes) and two genera (*Bacteroides* and *Faecalibacterium*) and decreasing the abundance of genus *Olsenella*. MMT also improved the intestinal epithelial barrier indicated by the up-regulated mRNA expression of *claudin-1*, *occludin*, and *ZO-1* and the increased length of microvilli in jejunum and the decreased levels of DAO and D-LA in serum. In addition, MMT enhanced the immune function indicated by the increased levels of immunoglobulins, the decreased levels of MPO and NO, the down-regulated mRNA expression of *IL-1β*, *IL-6*, and *TNF-α*, and the up-regulated mRNA expression of *IL-4* and *IL-10*. Moreover, MMT down-regulated the expression of jejunal TLRs/MAPK/NF-κB signaling pathway-related genes (*TLR2*, *TLR4*, *Myd88*, *TRAF6*, *NF-κB*, and *iNOS*) and related proteins (TRAF6, p38, ERK, NF-κB, and iNOS). In addition, MMT increased the antioxidant enzyme activities and the expression of Nrf2/HO-1 signaling pathway-related genes and thereby decreased the apoptosis-related genes expression. Spearman’s correlation analysis revealed that *Bacteroides*, *Faecalibacterium*, and *Olsenella* were related to the inflammatory index (MPO and NO), oxidative stress (T-AOC, T-SOD, and CAT) and intestinal integrity (D-LA and DAO). Taken together, MMT supplementation improved the growth performance of broilers by modulating intestinal microbiota, enhancing the intestinal barrier function, and improving inflammatory response, which might be mediated by inhibiting the TLRs/MAPK/NF-κB signaling pathway, and antioxidative capacity mediated by the Nrf2/HO-1 signaling pathway.

## 1. Introduction

The advent of antibiotics has profoundly impacted livestock health and welfare [1]. Antibiotics in feed have been shown to be effective in improving the overall performance and health of food animals, preventing various intestinal diseases, dropping morbidity and mortality, and raising feed utilization [2,3]. However, the misuse of antibiotics has caused severe consequences, such as antibiotic resistance (AMR), antibiotic residues in edible animal products, and environmental pollution of drugs, which seriously threaten the health of animals and humans [4]. Therefore, to deal with the unreasonable use of antibiotics, quite a few countries, including China, have restricted or prohibited antibiotics as growth promoters [5]. With the strict ban on antibiotics, several alternatives have been developed, including antimicrobial peptides, probiotics, antiviral molecules, and inorganic antimicrobial materials to maintain the health of livestock [6,7]. Among inorganic antimicrobial materials, montmorillonite has shown promise [8].

Montmorillonite (MT) is an aluminum silicate mineral clay with a huge surface area, strong absorptive capacity, and ion exchange capabilities, so it may be used to adsorb pathogenic bacteria and toxins [9]. Montmorillonite consists of two outer titans of silicon sandwiched between the inner aluminum layer, and exchangeable cations are quickly replaced by other cations between structural sheets [10]. Growing evidence has revealed its beneficial effects on animals [11]. The modified material is used to improve its actual performance by utilizing heat treatment, acid, alkali, and salt modification. Montmorillonite’s application as a metal ion carrier has attracted widespread interest. Some researchers have found that copper carried in montmorillonite has antibacterial effects [12]. Our research group uses ion-exchange reactions to synthesize copper-loaded montmorillonite (MMT). Supplementation with MMT could significantly improve the production performance and health of weaned piglets [10,13,14,15]. MMT showed a solid ability in poultry production due to reduce pathogenic bacteria colonization such as *Campylobacter jejuni* and *Clostridium perfringens* [11,16]. The possible mechanism is that MMT may aid adsorption of pathogenic bacteria, alter intestinal morphology, reduce intestinal permeability, and improve intestinal barrier functions of animals [17].

However, there is limited understanding of the interactions among MMT, the intestinal barrier, and the intestinal bacteria community. Therefore, this study was conducted to evaluate the impacts of MMT on growth performance, gut bacteria community, intestinal barrier function, and the underlying mechanism.

## 2. Materials and Methods

All experimental procedures and handling used in our work were approved (permission number: ZJU20160416) by the Animal Care and Use Committee of Zhejiang University (Hangzhou, China). The montmorillonite used in this experiment (>95% purity) was presented by Zhejiang Fenghong Biological Technology Co., Ltd. (Huzhou, Zhejiang, China). Modyfied montmorillonite (copper-loaded montmorillonite, the Cu concentration were 1.89 × 10^4^ mg/kg) was prepared by an ion-exchange reaction.

### 2.1. Animal Treatment and Experimental Design

Five hundred and forty Lingnan yellow broilers (Female) were purchased from ZhengDa Broilers Development Center of Zhejiang University (Hangzhou, China) and reared in XinXing Broiler Farm (Jiaxing, China). These 1-day-old chicks with similar weight were randomly assigned to three groups of six replicates of 30 birds per replicate. The treatments were as follows: (1) control group (CTR): broilers fed a basal diet; (2) MMT group: broilers fed a basal diet plus modified montmorillonite (500 mg/kg feed); (3) antibiotics group (ANTI): broilers fed the basal diet plus a combination of 250 mg/kg nosiheptide and 200 mg/kg oxytetracycline calciuma. The basal diet (Table 1) was formulated according to the nutrient requirements of broilers as recommended by the National Research Council (1994). Broilers were placed in an 18 h lighting device and the temperature of the room was kept at 32–34 °C for the first 3 d and then decreased by 2–3 °C weekly to a final temperature of 26 °C. Fresh water and feed was provided for broilers. This research was conducted according to the guidelines of the Animal Care and Use Committee of Zhejiang University.

### 2.2. Sample Collection and Treatment 

At the 63rd day of the experimental period, broilers were fasted for 4 h (05:00~9:00 a.m.) but provided with enough water. Six birds were randomly selected from each group and weighed. The broilers were then electro-stunned and scalded to collect tissue. The intestine was collected and rinsed with cold, sterile PBS at the same time to remove the attached impurities, and the mucosa of jejunum were gently scraped. 

### 2.3. Measurement of the Growth Performance

During the experiment, the feed intake for each repetition was recorded daily. The body weight of each pen was recorded on day 21, 38, and 63, respectively. The mortality rate was checked daily, and dead birds were recorded to adjust estimated value of ADG, ADFI, and feed conversion rate (F/R) appropriately. 

### 2.4. Serum-Biochemical Indexes

The serum immunoglobulin (IgG, IgM, and IgA) concentrations were determined by a Chicken specific ELISA kit (Nanjing Jiancheng, Bioengineering Institute, Nanjing, China). The total antioxidant capacity (T-AOC) and MDA, and the activity of total superoxide dismutase (T-SOD), glutathione peroxidase (GSH-Px), and catalase (CAT), D-lactate acid (D-LA), diamine oxidase (DAO), the activity of myeloperoxidase (MPO) and content of NO (nitrogen monoxide) were analyzed by a SpectraMax M5 (Molecular Devices, San Jose, CA, USA) using assay kits (Nanjing Jiancheng, Bioengineering Institute, Nanjing, China) according to the manufacturer’s instructions.

### 2.5. Transmission Electron Microscopy (TEM)

After fixation in 2.5% glutaraldehyde buffer, jejunum tissue was washed three times every fifteen minutes in 0.1 M cold phosphate buffer. The tissue was fixed in 0.1% osmium tetroxide (OsO4) cold buffer for 2 h, and then washed with phosphate buffer. After rapid dehydration in successively increasing ethanol solutions (30%, 50%, 70%, 95%, and 100%), the tissue was transferred to a 1:1 mixture of propylene oxide and epoxy-formaldehyde resin. After embedding, ultrathin sections (60–100 nm) were cut with an LKB Nova ultra-slicer (Leica Microsystems, Buffalo Grove, IL, USA) and stained with uranyl acetate. Electron microscopic images of gut mucosal cells and microvilli were taken by transmission electron microscope (JEOL, Tokyo, Japan) at 80 kV. The length of the microvilli was determined using the length measurement tool of ImageJ. 

### 2.6. RNA Extraction and RT-qPCR

The RNA was extracted using an RNAiso Plus Kit (Takara, Japan) following the instructions of the manufacturer. Purity and concentration of total RNA were determined using a nanodrop spectrophotometer (ND-2000, Thermo Fisher Scientific, Waltham, MA, USA). A reverse transcriptase M-MLV kit was used to perform reverse transcription of total RNA according to the instructions. qRT-PCR detection uses an ABI 7500 fluorescence detection system, and uses SYBR green (Takara, SYBR Premix Ex Taq TM II Kit) detection. Each sample was in triplicate and the primers are shown in Table 2. The 2^−^^∆∆ct^ method was used to calculate the relative quantification of all gene expressions, and β-actin was used as the internal reference gene. 

### 2.7. Western Blot Analysis

Lysis of jejunal tissue was prepared using lysis buffer (Sigma, Saint Louis, MO, USA). Total protein concentration was determined using the BCA method. Western blotting analysis was performed as described in a previous study [18]. The primary antibodies included rabbit anti-iNOS (HuaBio, Hangzhou, China), anti-ERK (Cell Signaling Technology, MA, USA), anti-phospho-ERK (Cell Signaling Technology, MA, USA), anti-p38 (Cell Signaling Technology, MA, USA), anti-phospho-p38 (Cell Signaling Technology, MA, USA), anti-JNK (Cell Signaling Technology, MA, USA), anti-phospho-JNK (Cell Signaling Technology, MA, USA), anti-NF-κB-p65 (Abcam, Cambridge, UK), anti-TRAF6 (HuaBio, Hangzhou, China), anti-HO1 (HuaBio, Hangzhou, China), and anti-GADPH (HuaBio, Hangzhou, China). The second antibody was HRP, goat anti-rabbit IgG, and goat anti-mouse IgG (HuaBio, Hangzhou, China). 

Protein bands were visualized with a chemiluminescence substrate (Millipore, MA, USA) and a gel-imaging system (Tanon Science and Technology, Shanghai) and analyzed with Image-J Analysis software (NIH, Bethesda, MD, USA). As an internal control, GADPH showed no difference among the three groups. In all cases, the density values of bands were corrected after subtracting background values. GAPDH was used as an internal reference protein. 

### 2.8. Intestinal Microbial DNA Extraction and High-Throughput Sequencing

Microbial genomic DNA was extracted from the cecal contents of 63-day-old broilers under sterile conditions using the TIANamp Stool DNA Kit (Tiangen, Beijing, China) according to the manufacturer’s protocol. The V3 to V4 region of the 16S rRNA gene was amplified using the 341F/805R primer pairs and sequencing was performed on an Illumina MiSeq platform (Illumina Inc., San Diego, CA, USA). Raw sequences were filtered by QIIME software (version 1.9.1) with 97% similarity and clustered into operational taxa (OTU). Alpha diversity, containing Shannon, Simpson, Ace, Chao1, and Coverage, was calculated to reflect the bacterial diversity and richness. Beta diversity on unweighted UniFrac was calculated based on OTU level. UniFrac-based principal coordinate analysis (PCoA) was performed to obtain principal coordinates and visualize from complex data. Differences in community structure between samples was calculated by nonmetric dimensional scaling (NMDS). The relative abundance of significant differences in phylum, class, order, and OTU levels was calculated by the one-way analysis of variance (ANOVA). Histogram of linear discriminant analysis (LDA) distribution was undertaken using LDA Effect Size Analysis (LEfSe) software. The 16S rRNA gene sequencing information was analyzed by PICRUSt to predict biological functions (EggNOG database) and metabolic pathways (KEGG database) of the bacterial community in broiler colonic content samples.

### 2.9. Statistical Analysis

Data were analyzed by one-way ANOVA followed by Turkey’s multiple comparison tests using SPSS software (version 20.0; IBM Inc., Armonk, NY, USA), and the results were expressed as mean ± standard error of the mean (SEM) or Standard Deviation (SD). Significant differences between means were declared at *p* < 0.05. The graphs were visualized using GraphPad prism 8.0 (GraphPad Software Inc., San Diego, CA, USA). 

## 3. Results

### 3.1. Growth Performance 

Compared with the CTR, MMT and ANTI notably (*p* < 0.05) increased the BW (21d, 38d, and 63d) and ADG (1-21d, 1-63d) and decreased the F:G of birds (Table 3). In addition, compared with MMT, dietary antibiotics supplementation obviously (*p* < 0.05) decreased the F:G (22-38d, 38-63d) of birds. However, there were no significant (*p* > 0.05) differences in BW (21d, 38d, 63d), ADG (1-21d, 22-38d, 38-63d, 1-63d), and F/G (1-21d, 1-63d) between the ANTI and MMT groups.

### 3.2. Cecum Microbiota Analysis

#### 3.2.1. Microbiota Diversity in Intestinal Contents

A Venn diagram illustrated that 517 OTUs overlapped in whole groups, while 531 OTUs were raised in CTR and ANTI, 545 OTUs appeared in CTR and MMT, and 526 OTUs overlapped in ANTI and MMT (Figure 1A). These results indicated that MMT and ANTI could regulate the diversity of intestinal microbiota to some extent. As shown in Figure 1B–F, ANTI tended to (0.05 < *p* < 0.1) decrease the microbial abundance (Simpson index) when compared with CTR. There were no significant differences (*p* > 0.05) among all groups for the indices of α diversity (the observed Shannon, ACE, and PD_whole_tree indices) of cecum. In addition, the weighted principal coordinate analysis (PCoA) and nonmetric multidimensional scaling (NMDS) analysis plots of cecal microbiota (Figure 1G,H) verified that there were obviously (*p* < 0.05) differences in microbial communities among all treatments, revealing that MMT and ANTI altered the intestinal bacterial community structure in the cecum.

#### 3.2.2. Cluster Analysis

Compared to the CTR, the relative abundance of Firmicutes and Bacteroidetes in MMT were significantly (*p* < 0.05) enhanced, but the relative abundance of *Actinobacteria* in MMT and ANTI was notably (*p* < 0.05) reduced (Figure 2C–E). As shown in Figure 2F–H, at genus level, an obvious reduction of *Olsenella* and an extraordinary rise of *Bacteroides* and *Phascolarctobacterium* in the MMT and ANTI were observed. Furthermore, the *Olsenella* level in MMT was obviously (*p* < 0.05) lower than that in ANTI. 

#### 3.2.3. Overall Structure Modulation of Gut Microbiota

Consistent with the bacterial changes described above, the cladogram generated from the linear discriminant analysis effect size (LEfSe) analysis, showed different intestinal microbiota compositions in all groups of broilers (Figure 3). The comparison of dominant bacterial taxa at the genus level suggested that MMT reduced the relative abundance of *Chroococcidiopsis-PCC-7203*, *Ruminococaaoeae-UCG-009*, and *Olsenella* compared with the CTR. Compared with cecal microbiota in control group, *Coriobacteriia*, *Coriobacteriales*, *Olsenella* and *Atopobiaceae* were decreased in ANTI group. In addition, MMT enhanced the *Gammaprotrobacteria* level and decrease the *Olsenella* and *Atopobiaceae* levels compared with ANTI.

#### 3.2.4. Predicted Metabolic Functions in the Gut Microbiota

Predicting the potential function of intestinal bacteria on nutrient metabolism in broilers after feeding on a diet with MMT and ANTI was undertook by analysis of biological functions and KEGG pathways by the PICRUSt program. The results showed that MMT and ANTI increased the abundance of microbiota related to the metabolism of cofactors and vitamins, and signal transduction and energy metabolism compared to the CTR (Figure 4). Dietary supplementation with MMT had greater effects on microbial metabolic functions than supplementation with ANTI in broilers.

### 3.3. Intestinal Physical Barrier Function

As shown in Figure 5A, TEM results showed that MMT and ANTI can significantly (*p* < 0.05) increase the length of microvilli compared with the CTR. Furthermore, broilers in the MMT and ANTI had significantly lower DAO and D-LA levels than that in the CTR group. MMT and ANTI also considerably (*p* < 0.05) increased the *claudin-1, occludin*, and *ZO-1* gene expression levels of jejunum mucosa in broilers. It had no significant difference (*p* > 0.05) in the content of DAO, D-LA, and the gene expression levels of *claudin-1, occludin*, and *ZO-1* between MMT and ANTI. These results suggested that MMT can improve intestinal barrier function in broilers.

### 3.4. Immune Responses and Inflammation

The serum levels of IgG and IgA in the MMT group were significantly higher than that in CTR and ANTI, while only the IgM level in ANTI was higher than that in the control group (Figure 6). MMT supplementation dramatically decreased the concentrations of NO and MPO compared with the control group. There were no obvious (*p* > 0.05) differences in the level of IgM, MPO, and NO between the MMT and ANTI groups. Figure 7 showed that the MMT and ANTI groups notably (*p* < 0.05) increased *IL-10* gene expression levels and down-regulated the *IL-1β, IL-6*, and *TNF-α* gene expression. Moreover, the gene expression level of *IL-4* in ANTI group was higher than that in MMT group.

MMT and ANTI decreased the TLRs/MAPK/NF-κB signal pathway-related gene expression (*TLR2, TLR4, Myd88, TRAF6*, and *NF-κB*). Western blot results also indicated that broilers in the CTR group had extraordinarily (*p* < 0.05) higher protein expression of phosphorylated p38 (p-p38-MAPK), p-ERK, iNOS, NF-κB, and TRAF6 levels compared with other groups (Figure 8). These results revealed that MMT has a solid capacity to modulate immunity and suppress inflammation mediated by inhibition of the TLRs/MAPK/NF-κB signaling pathway.

### 3.5. Antioxidant Status

For antioxidant status in the serum and jejunum mucosa, the T-AOC level and activity of T-SOD in the CTR were obviously (*p* < 0.05) lower than that in the other two groups. In addition, MMT and ANTI also considerably (*p* < 0.05) reduced the MDA content in the serum and jejunum mucosa compared with the CTR group (Table 4). GSH-Px activity was not affected by all treatments. Figure 9 showed that MMT and ANTI notably up-regulated gene expression levels of *CAT, SOD1, GPX4, HO-1, Nrf2*, and *keap1* compared with the CTR. No extraordinary differences in all antioxidant parameters were observed between the MMT and ANTI groups (*p* > 0.05). 

### 3.6. Apoptosis-Related Genes in the Jejunum

*Bax, Caspase-3*, and *Caspase-9* mRNA expression levels in the jejunal mucosa of the MMT and ANTI were lower than that in the CTR (*p* < 0.05). Moreover, MMT and ANTI increase the *Bcl2* (anti-apoptosis-related gene) mRNA expression level (Figure 10). 

### 3.7. Correlation Heat Map

Spearman correlation analysis was performed to further investigate specific relationships between phenotypic variables and the functional composition of microbial communities or microbial metabolism (Figure 11). As shown in Figure 12A, we found that inflammatory factors (MPO and NO), oxidative stress factor (MDA), and intestinal integrity (D-LA and DAO) were negatively related with the genera *Bacteroides and Faecalibacterium*, but positively related with the *Olsenella.* In addition, the serum immunoglobulin (IgG, IgM, and IgA) and antioxidant enzymes (T-AOC, T-SOD, CAT) were positively related with the genera *Bacteroides and Faecalibacterium*, but negatively related with the *Olsenella.* As shown in Figure 12B, our results indicated that serum immunoglobulin and antioxidase (T-AOC, T-SOD, CAT) were positively related with the pyruvate metabolism, two-component system, carbon metabolism, glycolysis/gluconeogenesis, and microbial metabolism in diverse environments, but negatively related with ribosome, aminoacyl-tRNA biosynthesis, and microbial metabolism in diverse environments and bio-synthesis amino acids. The results suggested that the promotion of gut health in broilers may be related to the altered structure of the cecal microbiota regulated by the addition of MMT.

## 4. Discussion

Prohibiting the use of antibiotic growth promoters has spurred the development of nutritional immunomodulators as an ideal strategy for maintaining intestinal health in today’s intensive poultry industry. Many studies have proved the utility of MMT as a potential alternative to antibiotics in developing performance and enhancing microbial microbiota in broilers [19,20,21,22,23]. Montmorillonite is extensively used as a feed additive, and its properties of improving the animal growth performance have attracted much attention [21,22,23]. Jiao reported that adding 2 g/kg montmorillonite significantly improved the pig weight gain, feed intake, and feed conversion [21]. In this study, the ADG, ADFI, and F:G were significantly changed in broilers after the supplementation of modified montmorillonite. Likewise, numerous studies have showed a significant improvement in body weight gain and feed conversion in pigs fed diets supplemented with MMT [24,25,26,27]. Depending on the adsorption of montmorillonite, feeding montmorillonite loaded with copper and zinc ions is beneficial to improve the growth performance of broilers [21,28]. Furthermore, the broilers body weight and F:G rate displayed no difference between the ANTI and the MMT groups. These findings demonstrated that MMT may serve as an efficient alternative to antibiotics and positively impact bird growth. 

A microbial barrier plays an essential role in intestinal functions, which includes nutrient absorption, intestinal mucosal barrier homeostasis, immune modulation, and pathogen defense in birds [29]. According to previous reports, MMT improved the gut barrier function, nutrient absorption and utilization, and production performance of poultry by regulating the gut microbiota [14,30]. This study also demonstrated that MMT could adjust gut microbiota. Although MMT did not alter the α-diversity index in the ileum of yellow-feathered broilers, the PCoA and NMDS analysis results proved an obvious difference in the composition of the microbial community between the CTR and the MMT, which may be the result of antibacterial activity of MMT. The positive antimicrobial activity of MMT may offer selective pressure to bacteria in birds. At the phylum level, *Firmicutes, Bacteroidetes*, and *Actinobacteria* were the main bacterial strains in our study, which is similar to previous studies [31,32]. Previous studies showed that *Firmicutes* and *Bacteroidetes* are associated with energy harvesting efficiency in poultry [33], and that the growth of broilers had a positive relationship with the *Firmicutes* and *Bacteroidetes* abundance and a negative relationship with *Actinobacteria* abundance [33,34]. Here, our results showed that MMT significantly raised the abundance of *Firmicutes* and *Bacteroidetes*, and reduced the abundance of *Actinobacteria*. In addition, it has been confirmed that a variety of members in *Bacteroidetes* and *Firmicutes* provide a beneficial role in host digestion in broilers [35]. The genus *Bacteroides* and *Phascolarctobacterium* display a positive effect on host health and disease resistance [36]. The decreased abundance of *Bacteroides* and *Phascolarctobacterium* was observed in patients with inflammatory bowel disease [36,37]. Our study also found that MMT increased the relative abundance of *Bacteroides* and *Phascolarctobacterium,* and decreased the relative abundance of *Olsenella,*
*Ruminococcaceae-UCG -009*. *Olsenella* is a genus belonging to the Actinobacteria phylum. Studies have shown that it is highly relevant to dysbiosis and inflammation [38,39]. *Ruminococcaceae-UCG -009* has been reported to be pathogenic bacteria when stimulated by stress, leading to an increased intestinal permeability, disrupted epithelial barriers, and intestinal diseases [40,41]. Furthermore, predicting the functional profile of the cecal microbial community using the PICRUSt program, we demonstrated that the addition of MMT enhanced the abundance of microbiota related to signal transduction, metabolism of cofactors and vitamins, energy metabolism, and replication and repair in the intestine. Clearly, the supplementation of MMT affected the dominant microbiota. Other studies have showed similar results that dietary MMT can help to maintain gut health by promoting the growth of beneficial bacteria [42]. 

Gut morphology can show the intestinal health status in animals [43]. Tight junctions between intestinal epithelial cells are critical for adjusting the permeability of the intestinal barrier and maintaining epithelial structural integrity. The disruption of the intestinal tight junctions will lead to the destruction of the gut barrier structure and increased gut permeability [44]. The serum D-LA and DAO concentrations could be used as biomarkers of gut barrier integrity [45], which are enhanced in birds when the intestinal barrier is compromised [46]. This study indicated that MMT and ANTI could obviously (*p* < 0.05) increase the microvilli length, and improved jejunum *claudin-1*, *occludin*, and *ZO-1* mRNA expression levels, which is consistent with the decreased serum levels of DAO and D-LA in the MMT group. Consistently, the regulation and repair of the gut barrier by MMT have been demonstrated by numerous studies as it can modulate the expression of tight junction proteins [47,48]. In addition, we found that there was no obvious differences in serum D-LA and DAO concentrations, and expression levels of *occludin, claudins,* and *ZO-1* between the MMT and ANTI groups, revealing that MMT may serve as a good alternative to antibiotics to alleviate intestinal mucosal permeability, which is in keeping with the state of intestinal health in broilers [21]. These findings suggested that MMT could improve the gut barrier function in broilers and promote intestinal epidermal cell metabolism [21,25].

The immune system protects the body from foreign substances and protects invasion by pathogens [49]. In broilers, immunoglobulins including IgA, IgG, and IgM are the important indicators of immune system status [50]. Cytokine secretion is often critical for triggering the innate defense program and then regulating the immune response of an adaptive system [51]. The TLR family can recognize the unique structural components of fungi, viruses, and bacteria to activate the inflammatory response [52]. Motivation of the extracellular domain of TLR triggers the intracellular binding of MyD88 to its cytoplasmic domain, which activates NF-κB; the activated NF-κB then promotes its subunits, such as NF-κB p65, to translocate to the nucleus and turn on cytokine gene expression, ultimately leading to the release of numerous pro-inflammatory mediators, including IL-1β and TNF-α [53]. Our study showed that serum levels of IgG, IgM, and IgA were significantly enhanced by MMT and ANTI treatment, which was in agreement with the results of other reports [30]. In addition, we found that the mRNA expression levels of *TLR2*, *TLR4*, *MyD88*, *TRAF6*, *NF-kB p65*, *iNOS*, *IL-1β*, *IL-8*, and *TNF-α* in the intestine of MMT group were decreased, indicating that MMT may inhibit the release of pro-inflammatory cytokines by regulating signal pathways TLRs/MAPK/NF-κB. Similarly, several studies have shown that mRNA levels of intestinal mucosa pro-inflammatory mediators can be downregulated by feeding with MMT [14,30]. Pathogens such as *E. coli* and *Salmonella* are known to be recognized by TLRs, especially TLR4, which stimulates downstream activation of signaling cascades that control pro-inflammatory responses. The previous study also confirmed that dietary MMT reduces the number of *E. coli*, *Salmonella*, and *Clostridium* in the small intestine of laying hens [14,30]. Therefore, it suggests that the suppression of pro-inflammatory cytokines may be attributed to the improvement of the intestinal environment and the suppression of pathogen populations. In addition, the downregulation of *MyD88*, *TRAF6*, and *NF-κB* mRNA expression in the jejunum indicates that dietary MMT could decrease inflammation response in broilers. Given these results, MMT could improve anti-inflammatory response by inhibiting TLRs/MAPK/NF-κB pathway.

The body’s antioxidant function is an essential part of preserving the intestinal barrier integrity and preventing pathogen infection [14,54]. The results of this study showed that MMT can reduce lipid peroxidation indicated by the lower MDA level, and strengthen the antioxidant capacities of broilers indicated by the increased activities of T-SOD and CAT and T-AOC level. Our findings were supported by the work of Chen et al. [14]. The keap1-Nrf2 signaling pathway is an effective way for the body to relieve stress by encoding antioxidant enzymes and phase II detoxification enzymes to resist endogenous or exogenous oxidative stress [55,56,57,58]. Here, an obvious increase in *Nrf2*, *keap1*, and *HO1* gene expression was noted in the MMT group, which suggested that the keap1-Nrf2 pathway is involved in an antioxidized capacity exerted by MMT. We also found that key apoptosis-related genes (such as Bax, caspase-3, and caspase-9) were significantly decreased in the MMT group, proving that MMT has good anti-apoptosis properties. Oxidative stress and inflammation generally bring about apoptosis in vivo or in vitro [59]. The above results demonstrated that MMT contributes to reducing apoptosis of intestinal epithelial cells by enhancing the antioxidant and anti-inflammatory function.

In this study, Spearman’s correlation analysis revealed there were strong correlations among the phenotypic variables, microbial communities, and microbial metabolic functions, such as the increased *Bacteroides* in the MMT group was highly negative correlation with MPO, MDA, DAO, and D-LA, and remarkably positive link with IgG, IgM, and IgA. However, decreased *Olsenella* in MMT group was positively correlated with MDA, MPO, NO, DAO, and D-LA, and negatively correlated with IgG, IgA, IgM, T-SOD, T-AOC, and CAT. Consistently, it was reported that *Bacteroides* could significantly reduce the fasting hyperglycemia and high plasma concentrations of proinflammatory cytokines in mice [60], and exhibited a marked anti-inflammatory activity against *E.coli*-induced IL-8 release, and showing that an increased abundance of *Bacteroides* might suppress inflammatory responses [61,62]. Furthermore, the antioxidant and immune functions were positively correlated with the pyruvate metabolism, ribosome, glycolysis/gluconeogenesis, and pyrimidine metabolism of microbial communities, but negatively with metabolic pathways, microbial metabolism, and carbon metabolism of microbial communities. These findings suggested that MMT improved the growth performance of broilers, which may be associated with the altered gut microbiota and bacterial metabolic functions, which should be further confirmed by whole shotgun metagenomic sequencing due to the limited taxonomic and functional attributes provided by 16S rRNA gene sequencing.

## 5. Conclusions

In summary, supplemental MMT can improve the growth performance in early period (1-21d) and intestinal villi structure by modulating the composition of gut bacteria, improving the mucosal anti-inflammatory response mediated by the TLRs/MAPK/NF-κB signaling pathway, and antioxidative capacity mediated by the keap1-Nrf2 pathway (Figure 13). The findings demonstrate the potential of MMT to partly substitute antibiotic usage in yellow-feathered broiler production.

## Figures and Tables

**Figure 1 antioxidants-11-01799-f001:**
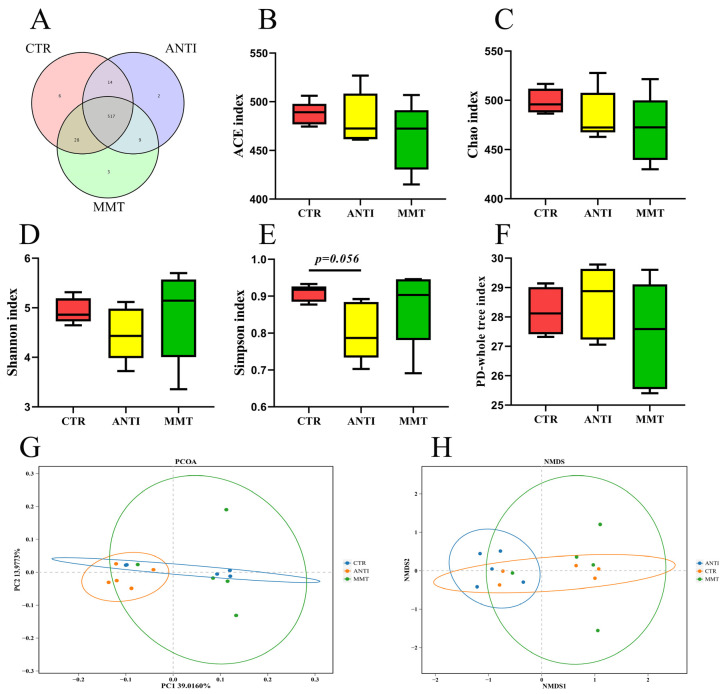
Effect of MMT on gut microbiota diversities of broilers: (**A**) Venn diagram; (**B**) ACE index; (**C**) Chao1 index; (**D**) Shannon index; (**E**) Simpson index; (**F**) PD-whole tree; (**G**) Principal coordinate analysis (PCoA)analysis of weighted UniFrac distance; (**H**) NMDS analysis of weighted UniFrac distance. Data are presented as means ± SD (n = 5).

**Figure 2 antioxidants-11-01799-f002:**
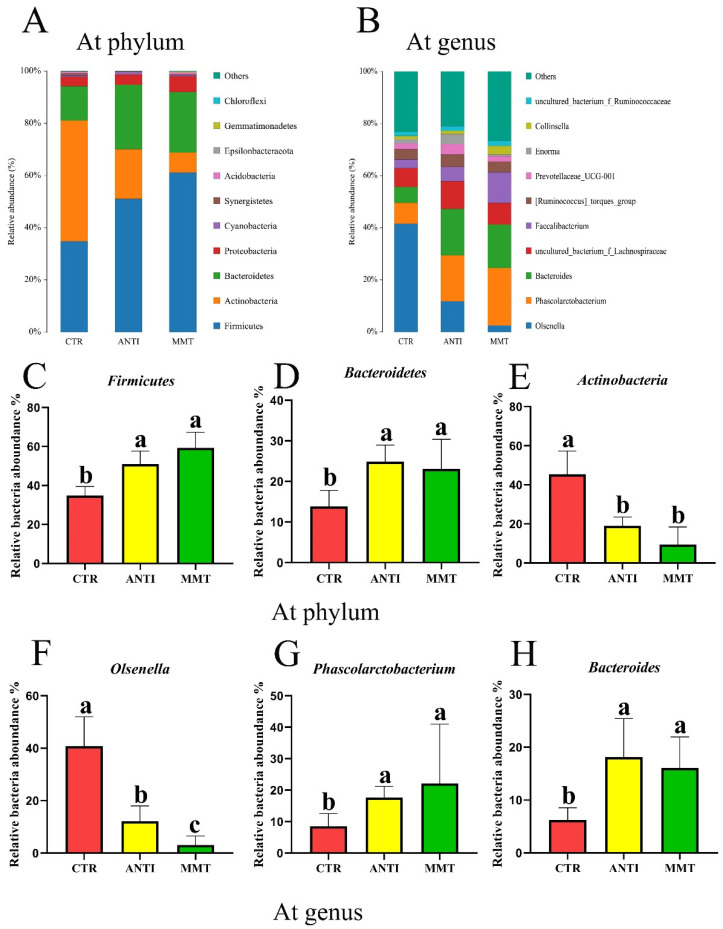
Bacterial taxonomic composition of cecum contents: (**A**) relative abundance of phylum level; (**B**) relative abundance of genus level; (**C**–**E**) the relative abundance of significant differential bacteria on phylum; (**F**–**H**) the relative abundance of significant differential bacteria on genus. Data are presented as means ± SD (n = 6). ^a,b,c^ Means within a row with different superscripts differ significantly (*p* < 0.05).

**Figure 3 antioxidants-11-01799-f003:**
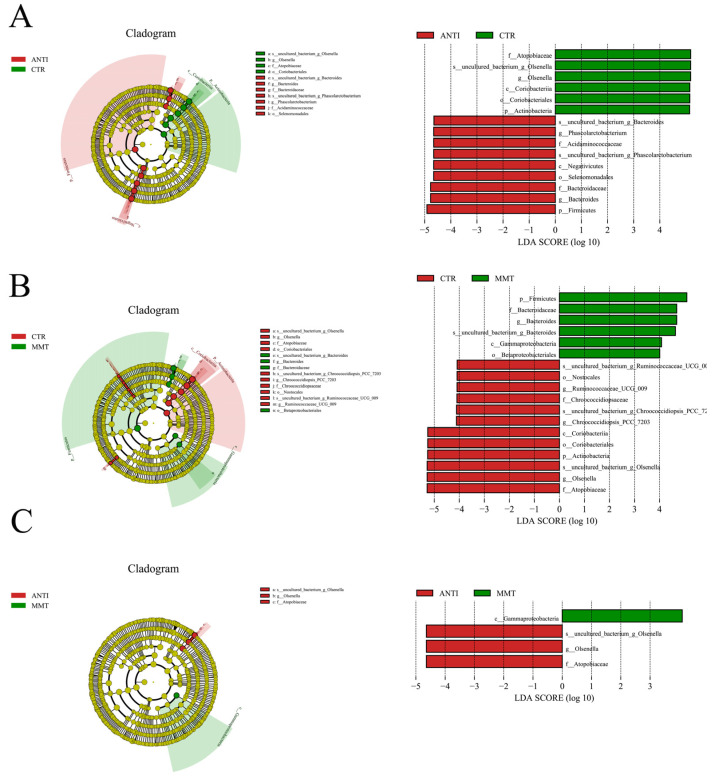
LEfSe cladogram and LEfSe bar: (**A**) CTR vs. ANTI; (**B**) CTR vs. MMT; (**C**) ANTI vs. MMT.

**Figure 4 antioxidants-11-01799-f004:**
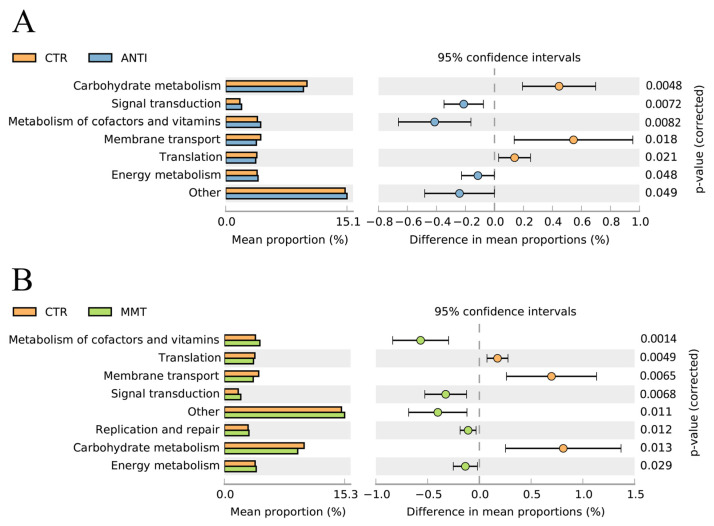
Comparison to predicted metabolic pathway abundance between the groups by statistical analysis of taxonomic and functional profiles (STAMP) at Level 2: (**A**) CTR vs. ANTI; (**B**) CTR vs. MMT.

**Figure 5 antioxidants-11-01799-f005:**
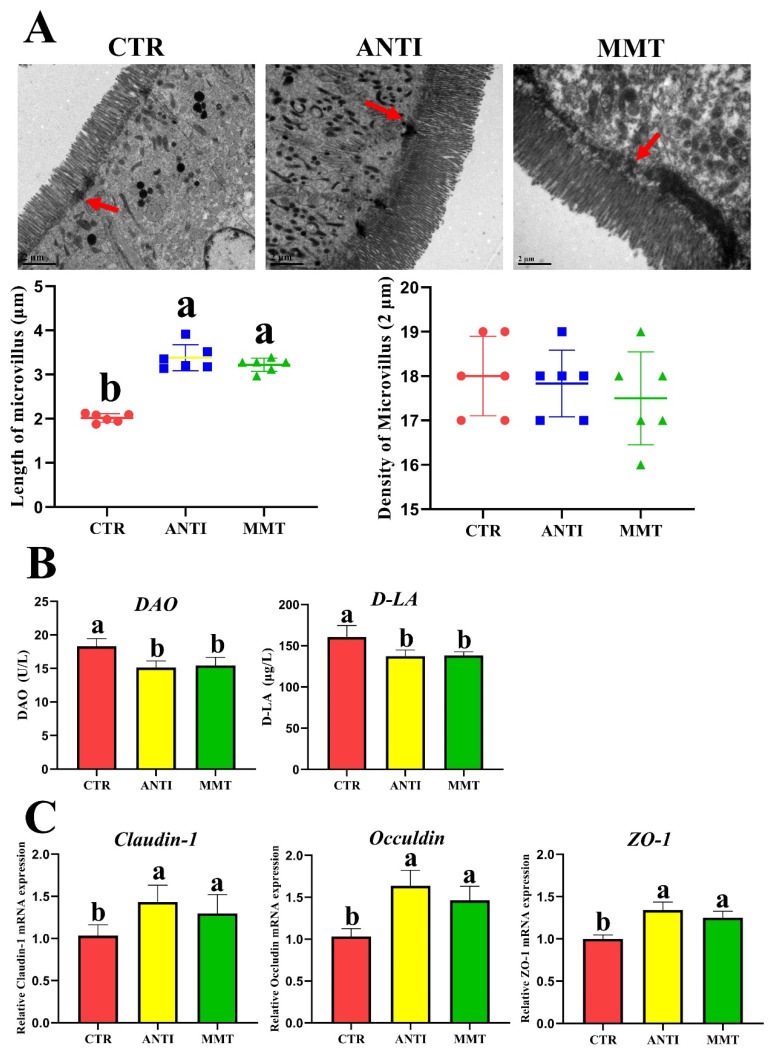
Effects of dietary supplementation with MMT on the intestinal physical barrier function: (**A**) transmission electron micrographs of the jejunum microvilli in broilers, the red arrow indicates desmosomes (DS); (**B**) DAO and D-LA activity in serum; (**C**) the jejunum relative mRNA expressions of *claudin-1*, *occludin*, and *ZO-1* were analyzed by real-time qPCR. ^a,b^ Data are presented as means ± SD (n = 6). ZO-1, zonula occludens 1.

**Figure 6 antioxidants-11-01799-f006:**
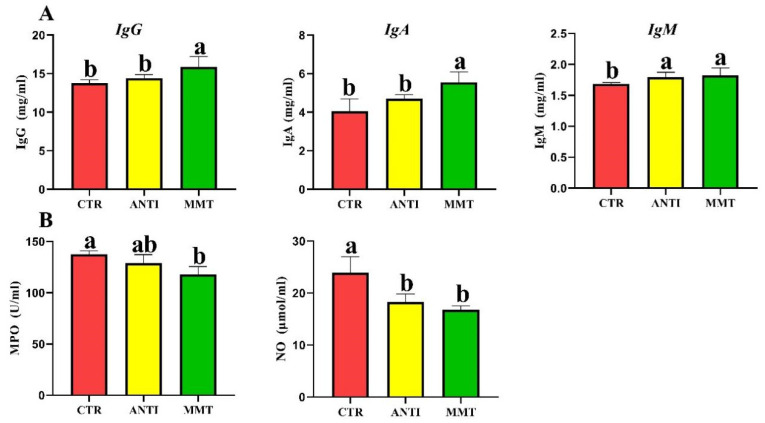
Effects of dietary supplementation with MMT on the content of serum immunity index in broilers: (**A**) content of IgG, IgA, and IgM in serum; (**B**) content of MPO and NO in serum. Data are presented as means ± SD (n = 6). ^a,b^ Means within a row with different superscripts differ significantly (*p* < 0.05).

**Figure 7 antioxidants-11-01799-f007:**
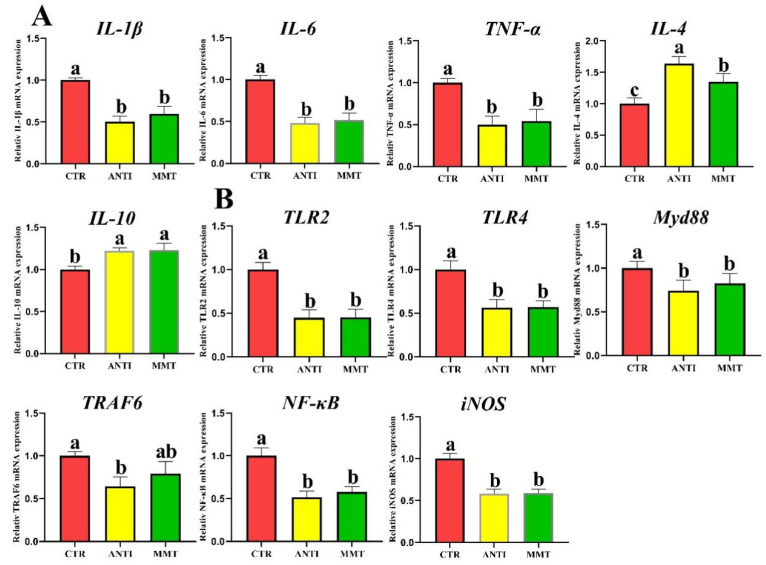
Effects of dietary supplementation with MMT on the relative mRNA expression of cytokine genes and inflammation-related pathways in jejunum: (**A**) the relative mRNA expressions of cytokines *(IL-1**β, IL-6, TNF-**α, IL-4, IL-10*); (**B**) the relative mRNA expressions of inflammation-related pathways (*TLR2, TLR4, Myd88, TRAF6, NF-**κB, iNOS*). Data are presented as means ± SD (n = 6). ^a,b,c^ Means within a row with different superscripts differ significantly (*p* < 0.05).

**Figure 8 antioxidants-11-01799-f008:**
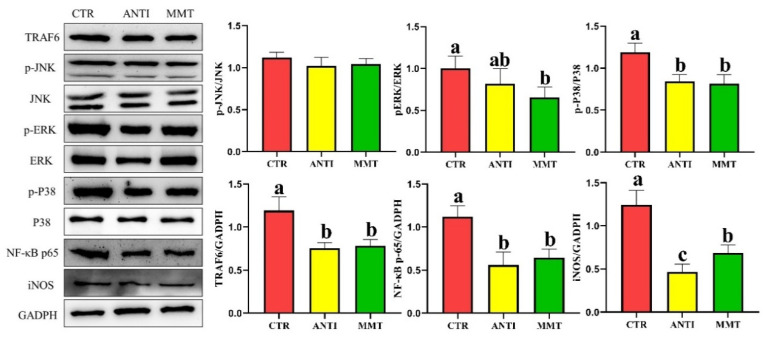
Effects of dietary supplementation with MMT on the relative protein expression of inflammation-related pathways in jejunum. Data are presented as means ± SD (n = 3). ^a,b,c^ Means within a row with different superscripts differ significantly (*p* < 0.05).

**Figure 9 antioxidants-11-01799-f009:**
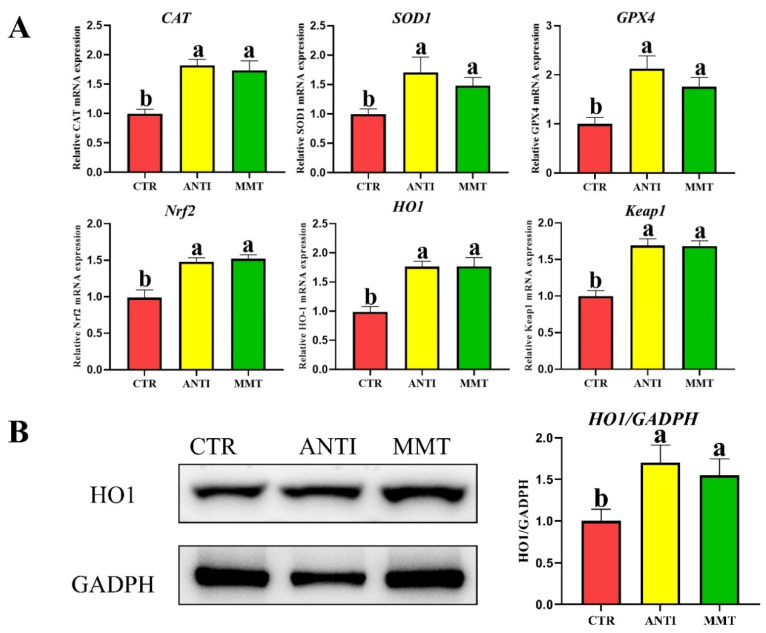
Effects of dietary supplementation with MMT on the relative mRNA and protein expression of antioxidative mediators’ genes in jejunum. (**A**): the relative mRNA of antioxidative mediators’ genes in jejunum. (**B**): the relative protein expression of antioxidative in jejunum. Data are presented as means ± SD (n = 6). ^a,b^ Means within a row with different superscripts differ significantly (*p* < 0.05).

**Figure 10 antioxidants-11-01799-f010:**
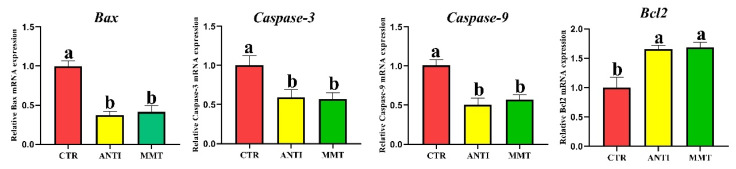
Effects of dietary supplementation with MMT on the relative mRNA expression of apoptosis-related genes. Data are presented as means ± SD (n = 6). ^a,b^ Means within a row with different superscripts differ significantly (*p* < 0.05).

**Figure 11 antioxidants-11-01799-f011:**
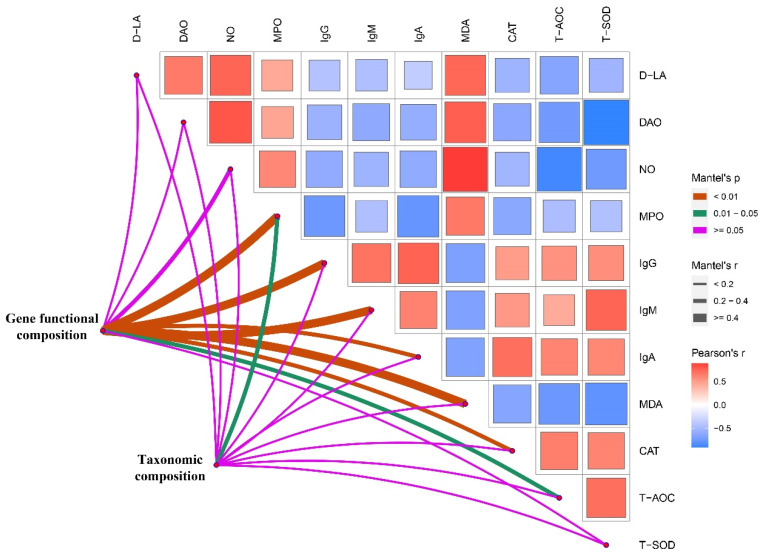
Relationships among phenotypic variables, the microbial community, and metabolic functional composition. Pairwise comparisons of phenotypic variables with a color gradient denoting Pearson correlation coefficient. Taxonomic and functional community structures were related to each phenotypic variable by Mantel correlation (based on Bray–Curtis dissimilarity). The edge width represents the Mantel’s r statistic for the corresponding distance correlations, and the edge color denotes the statistical significance.

**Figure 12 antioxidants-11-01799-f012:**
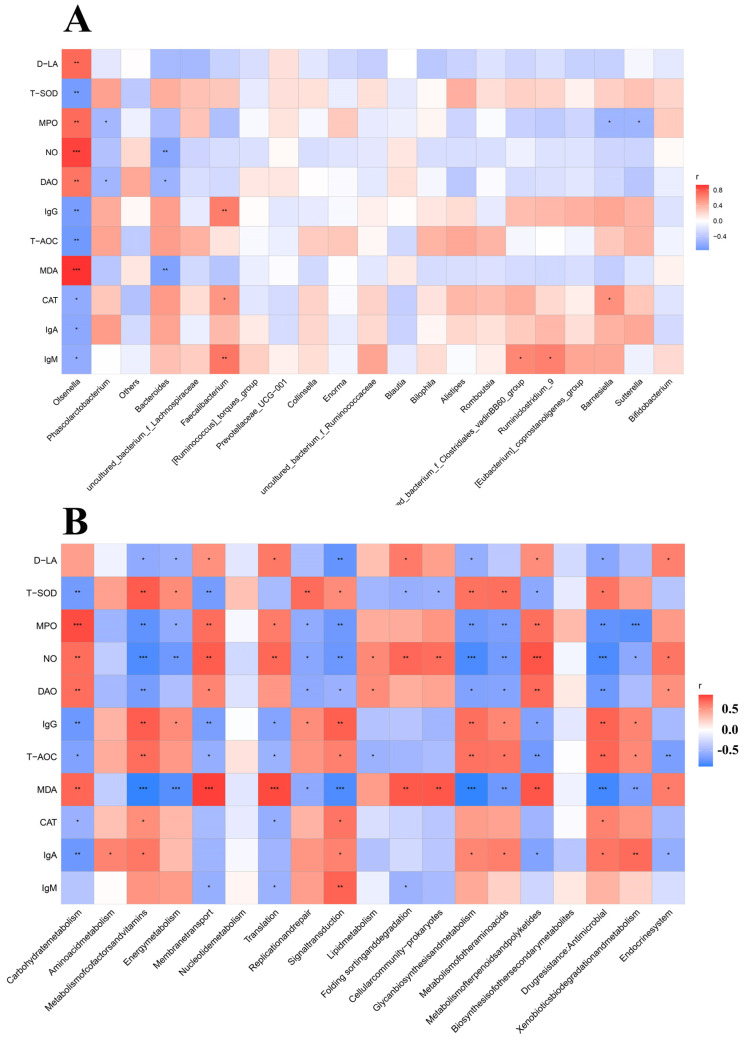
Pearson’s correlation analysis. Correlation of phenotypic variables and microbial communities (**A**) and microbial-predicted metabolic pathway functions (**B**). The color and the dot size represent the correlation coefficient. * *p* < 0.05, ** *p* < 0.01, *** *p* < 0.001.

**Figure 13 antioxidants-11-01799-f013:**
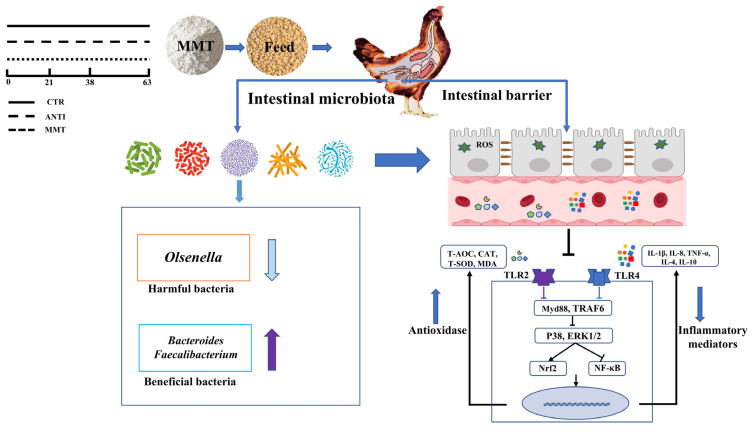
MMT supplementation improved the growth performance of broilers by modulating intestinal microbiota, enhancing the intestinal barrier function, and improving inflammatory response, which might be mediated by inhibiting the TLRs/MAPK/NF-κB signaling pathway, and antioxi-dative capacity mediated by the Nrf2/HO-1 signaling pathway. ↑ Arrows indicate the effect of stimulation; ↓ Arrows indicate the effect of suppression.

**Table 1 antioxidants-11-01799-t001:** Formulation and calculated composition of the basal diet (as-fed basis).

Item (%, Unless Otherwise Indicated)	Amount
Ingredients	
Corn	60.00
Soybean meal	28.50
Fish meal	2.00
Wheat middling	4.50
Dicalcium phosphate	1.30
Limestone	2.25
50% choline chloride	0.15
Salt	0.30
Vitamin and mineral premix 1	1.00
Total	100.00
Calculated Nutrient Level	
CP (crude protein)	22.39
Total P	0.70
Ca (calcium)	1.00
Lys (Lysine)	1.17
Met + Cys (Methionine + Cysteine)	0.65
Met (Methionine)	0.48
ME (MJ/kg)	12.22

The premix provides the following for per kilogram diet: vitamin A 12,000 IU, vitamin D_3_ 3000 IU, vitamin E 25 IU, vitamin K_3_ 22 mg, vitamin B_1_ 3 mg, vitamin B_2_ 8 mg, vitamin B_6_ 4 mg, vitamin B12 0.02 mg, nicotinic acid 45 mg, pantothenic acid 12.5 mg, biotin 0.1 mg, folic acid 1 mg, copper 8 mg, iron 80 mg, zinc 60 mg, manganese 100 mg, selenium 0.15 mg, iodine 0.35 mg. Nutrient level: lysine and methionine were calculated values, whereas others were analyzed values.

**Table 2 antioxidants-11-01799-t002:** Sequences of the oligonucleotide primers used for quantitative real-time PCR ^1^.

Gene Name	Accession Number	Sequence (5′-3′)
*β-actin*	NM_205518.1	F: CATTGTCCACCGCAAATGCT
		R: AAGCCATGCCAATCTCGTCT
*TLR2*	NM_001161650.2	F: GGATTTCTCGCACTTTCGCC
		R: AAAGGGACAAAAAGGCAATGGA
*TLR4*	NM_001030693.1	F: GGCTCAACCTCACGTTGGTA
		R: AGTCCGTTCTGAAATGCCGT
*TRAF6*	XM_046942060.1	F: CGTCGTGCATTCCTACCGAT
		R: GACAGCATAGGGCAACGAGT
*Myd88*	NM_010851.3	F: AGGCATCACCACCCTTGATG
		R: CGAAAAGTTCCGGCGTTTGT
*NF-κB*	NM_205134.1	F: TGGAGAAGGCTATGCAGCTT
		R: CATCCTGGACAGCAGTGAGA
*iNOS*	NM_204961.2	F: GAACAGCCAGCTCATCCGATA
		R: CCCAAGCTCAATGCACAACTT
*IL-1β*	NM_204524.1	F: TCTGCCTGCAGAAGAAGCC
		R: CCTCACTTTCTGGCTGGAGG
*IL-6*	NM_204628.1	F: CCTTTCAGACCTACCTGGAATT
		R: ACTTCATCGGGATTTATCACCA
*TNF-α*	NM_205427.1	F: CAACGACACCATCCTGGACA
		R: ATCCGGTTGAGGAGGCTTTG
*IL-4*	NM_001007079.1	F: GTGCCCACGCTGTGCTTAC
		R: AGGAAACCTCTCCCTGGATGTC
*IL-10*	NM_001004414.4	F: CGCTGTCACCGCTTCTTCA
		R: TCCCGTTCTCATCCATCTTCTC
*Occludin*	XM_025144247.1	F: CGGAGCCCAGACTACCAAAG
		R: TTACACAGCTTCAGCCTTACA
*Claudin-1*	NM_001013611.2	F: GGTATGGCAACAGAGTGGCT
		R: CAGCCAATGAAGAGGGCTGA
*ZO-1*	NM_204918.1	F: TAAAATGGACAGGCGCTGACA
		R: TTGGGCGTGACGTATAGCTG
*CAT*	NM_214301.2	F: TCCAGCCAGTGACCAGATGA
		R: CTCTCCCGGTCAAAGTGAGC
*SOD1*	NM_001190422.1	F: CAGGGCACCATCTACTTCGAGC
		R: ACGTGCCTCTCTTGATCCTTTG
*GPX4*	NM_214407.1	F: TGGATGAAAGTCCAGCCCAAG
		R: CTAGAGGTAGCACGGCAGGT
*Nrf2*	NM_001004027	F: CGCTCCCGAATGAACAC
		R: GCTCCTGCACCTCCTC
*HO1*	XM_003133500.5	F: GCCCCTGGAAGCGTTAAAC
		R: GGACTGTATCCCCAGAAGGTTGT
*Keap1*	NM_001114671.1	F: GCGTTCCGCACCTCCC
		R: GACAAGGGACAACACCACCA
*Bax*	XM_015274882.1	F: GTGATGGCATGGGACATAGCTC
		R: TGGCGTAGACCTTGCGGATAA
*Bcl2*	NM_205339.2	F: GATGACCGAGTACCTGAACC
		R: CAGGAGAAATCGAACAAAGGC
*Caspase-3*	NM_204725.1	F: ACTCTGGAAATTCTGCCTGATGAC
		R: CATCTGCATCCGTGCCTGA
*Caspase-9*	XM_424580.6	F: TCAGACATCGTATCCTCCA
		R: AAGTCACAGCAGGGACA

^1^ Note: TLR2: toll-like receptor 2, TLR4: toll-like receptor 4, TRAF6: TNF receptor associated factor 6, Myd88: myeloid differentiation factor 88, NF-κB: nuclear factor kappa-B, iNOS: inducible nitric oxide synthase, IL-1β: interleukin-1β, IL-6: interleukin-6, TNF-α: tumor necrosis factor-α, IL-4: interleukin-4, IL-10: interleukin-10, ZO-1: zonula occludens, CAT: catalase, SOD1: superoxide dismutase 1, GPX4: glutathione peroxidase 4, Nrf2: nuclear factor erythroid 2-related factor 2, HO1: recombinant heme oxygenase 1, Bax: recombinant Bcl2 associated X protein, Bcl2: B-cell lymphoma-2.

**Table 3 antioxidants-11-01799-t003:** Effect of MMT on the growth performance of yellow-feathered broilers.

Stage	Indicator	CTR	ANTI	MMT	SEM	*p*-Value
**BW**	**1d**	37.974	37.908	38.434	0.337	0.285
	**21d**	411.842 ^b^	429.883 ^a^	434.000 ^a^	5.087	0.004
	**38d**	1083.445 ^b^	1141.136 ^a^	1125.938 ^ab^	19.987	0.045
	**63d**	1928.834 ^b^	2036.510 ^a^	2026.644 ^a^	33.738	0.020
**1-21d**	**ADG**	17.803 ^b^	18.666 ^a^	18.837 ^a^	0.238	0.004
	**ADFI**	30.502	30.712	31.169	0.590	0.537
	**F/G**	1.714	1.647	1.655	0.047	0.346
**22-38d**	**ADG**	39.506	41.838	40.702	1.146	0.182
	**ADFI**	81.376	80.519	82.792	1.746	0.453
	**F/G**	2.060 ^a^	1.927 ^b^	2.034 ^a^	0.043	0.030
**38-63d**	**ADG**	33.816	35.815	36.028	1.784	0.427
	**ADFI**	93.329 ^b^	91.096 ^b^	98.466 ^a^	2.399	0.041
	**F/G**	2.760 ^a^	2.552 ^b^	2.741 ^a^	0.085	0.067
**1-63d**	**ADG**	30.014 ^b^	31.724 ^a^	31.559 ^a^	0.536	0.020
	**ADFI**	71.736	71.609	74.142	1.383	0.175
	**F/G**	2.390 ^a^	2.258 ^b^	2.350 ^ab^	0.045	0.041

^a,b^ Means within a row with different superscripts differ significantly (*p* < 0.05). Means represent 6 replicates per treatment, with 15 broilers per replicate. F/G: feed conversion ratio.

**Table 4 antioxidants-11-01799-t004:** Serum and jejunum mucosa antioxidant status of broiler ^1^.

Item	CTR	ANTI	MMT	SEM	*p*-Value
Serum					
T-AOC (U/mL)	7.16 ^b^	9.01 ^a^	9.57 ^a^	0.488	0.001
T-SOD (U/mL)	129.17 ^b^	148.42 ^a^	148.89 ^a^	3.875	0.001
CAT (U/mL)	6.09 ^b^	7.54 ^a^	7.78 ^a^	0.462	0.005
GSH-Px (U/mL)	2045.56	2138.04	2035.74	107.451	0.587
MDA (nmol/mL)	7.94 ^a^	6.84 ^b^	6.78 ^b^	0.177	<0.001
Jejunal mucosa					
T-AOC (U/mg prot)	2.64 ^b^	3.48 ^a^	3.61 ^a^	0.152	0.001
T-SOD (U/mg prot)	130.6 ^b^	154.03 ^a^	155.49 ^a^	3.291	0.001
CAT (U/mg prot)	13.33	13.4	13.14	0.346	0.743
GSH-Px (U/mg prot)	1404.92	1407.07	1408.67	7.509	0.883
MDA (nmol/mg prot)	3.29 ^a^	2.66 ^b^	2.61 ^b^	0.11	0.001

^1^ Results are the means of each group of 6 laying hens. T-AOC, total antioxidant capacity; SOD, superoxide dismutase; CAT, catalase; GSH-Px glutathione peroxidase; MDA, malondialdehyde. ^a,b^ Value differences in the same row differ significantly (*p* <  0.05).

## Data Availability

Data is contained within the article.

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
