# Peer review of "Modified Montmorillonite Improved Growth Performance of Broilers by Modulating Intestinal Microbiota and Enhancing Intestinal Barriers, Anti-Inflammatory Response, and Antioxidative Capacity"

_antioxidants, 2022, doi:10.3390/antiox11091799_

Round 1
Reviewer 1 Report
This research is interesting and a lot of work has been done. However, It must be significantly improved to be recommended for publication.
Specific remarks;
1. Line 84. What was the sex of birds (male/female/mixed)? This is essential to evaluate birds performance.
2. Line 100. Why You based your diets on NRC (1994) – there are much more up-to-date nutritional guidelines for currently reared lines of broilers.
3. Line 102. The reached rearing temperature of 26ËšC – is that ok for this birds? It seems to high, which is associated with the heat stress and antioxidant system as a consequence…
4. Line 115. What does mean that broilers after stunning were mixed?
5. Overall, the material and method section should be written in past tense.
6. l. 142. So how indeed quantitative tight junctions were done in this study?
7. Table 2. Provide full names for gene abbreviations. Plus did you measure expression efficiency of reference gene? How did you chose this gene to be used as a reference?
8. Statistical analysis – what kind of test did you apply for means separation?
9. Lines 196-203. Comparing to CTR, ANTI DID NOT significantly influenced BW at 38 day, the same was true for F/G.
10. Figure 1. Why the number of replicates (n=) was 5 if 6 birds was sampled? Provide exact number of n= for each analysis.
11. Lines 322-324. Move this comment or similar given previously to the discussion section or prepare mixed section.
12. General remark to the conclusion;
It has been drawn not strictly to the results obtained (i) dietary treatment improved/affected significantly only selected indices like BW and over a selected period of birds growth, (ii) you did not measure herein inflammatory response to stress factors and (iii) you investigate only chicken response thus you can not extrapolate for other species.
Author Response
This research is interesting and a lot of work has been done. However, It must be significantly improved to be recommended for publication.
Response: We gratefully appreciate the reviewer’s positive and professional comments.
Major comments
Comment 1. Line 84. What was the sex of birds (male/female/mixed)? This is essential to evaluate birds performance.
Response: We deeply appreciate the reviewer’s professional comments. Male Lingnan yellow broilers was selected in the present study, and detailed information about the sex of birds had been added in MATERIALS AND METHODS part.
Comment 2. Line 100. Why You based your diets on NRC (1994) - there are much more up-to-date nutritional guidelines for currently reared lines of broilers.
Response: We deeply appreciate the reviewer’s professional comments. We raise yellow feather broilers, a Chinese local breed. Its nutritional requirements are close to NRC1994, which has been confirmed in previous studies [1-6].
Reference:
[1] Selaledi, L.; Baloyi, J.; Mbajiorgu, C.; et al. Meat Quality Parameters of Boschveld Indigenous Chickens as Influenced by Dietary Yellow Mealworm Meal. Foods 2021, 10.
[2] Song, W.J.; Song, Q.L.; Chen, X.L.; et al. Effects of honeycomb extract on the growth performance, carcass traits, immunity, antioxidant function and intestinal microorganisms of yellow bantam broilers. Poultry science 2022, 101, 101811-101811.
[3] Chen, S.; Xue, Y.; Shen, Y.; et al. Effects of different selenium sources on duodenum and jejunum tight junction network and growth performance of broilers in a model of fluorine-induced chronic oxidative stress. Poultry Science 2022, 101.
[4] Liao, S.; Liao, L.; Huang, P.; et al. Effects of Different Levels of Garlic Straw Powder on Growth Performance, Meat Quality, Antioxidant and Intestinal Mucosal Morphology of Yellow-Feathered Broilers. Frontiers in Physiology 2022, 13.
[5] Xue, J.; Fang, C.; Mu, R.; et al. Potential Mechanism and Effects of Different Selenium Sources and Different Effective Microorganism Supplementation Levels on Growth Performance, Meat Quality, and Muscle Fiber Characteristics of Three-Yellow Chickens. Frontiers in Nutrition 2022, 9.
[6] Deng, S.; Xing, T.; Li, C.; et al. The Effect of Breed and Age on the Growth Performance, Carcass Traits and Metabolic Profile in Breast Muscle of Chinese Indigenous Chickens. Foods 2022, 11.
Comment 3. Line 102. The reached rearing temperature of 26ËšC – is that ok for this birds? It seems to high, which is associated with the heat stress and antioxidant system as a consequence…
Response: Many thanks for the reviewer’s professional comments. According to the Chinese yellow-feathered Broiler Management Handbook, the temperature of the room was kept at 32 to 34°C for the first 3 day and then decreased by 2 to 3°C per week for 3 weeks to a final temperature of 26°C, which we strictly followed the yellow-feathered broiler production procedure in this study. “Place broilers in a 18-hour lighting device and the temperature of the room was kept at 32-34°C for the first 3 d and then decreased by 2-3 °C weekly to a final temperature of 26 °C.”
Comment 4. Line 115. What does mean that broilers after stunning were mixed?
Response: Sorry for the inconvenience in the process of your review and we had revised this sentence. “Six birds were randomly selected from each group and weighed. The broilers were then electro-stunned and scalded to collect tissue.”
Comment 5. Overall, the material and method section should be written in past tense.
Response: Done as requested. Many thanks for the reviewer’s kind advice.
Comment 6. l.142. So how indeed quantitative tight junctions were done in this study?
Response: Sorry for the mistake we made and we had deleted this sentence.
Comment 7. Table 2. Provide full names for gene abbreviations. Plus did you measure expression efficiency of reference gene? How did you chose this gene to be used as a reference?
Response: Many thanks for your kind advice and we had added the full names for gene abbreviations in Table 2. β-actin is one of the two nonmuscle cytoskeletal actins and can be stably expressed in mucosa tissues[1], which is commonly used as internal reference (housekeeping gene) for qPCR[2-4].
Reference:
[1]Shim, J.; Shim, E.; Kim, G.H.; et al. Keeping house: evaluation of housekeeping genes for real-time PCR in the red alga, Bostrychia moritziana (Florideophyceae). Algae 2016, 31, 167-174.
[2] Roy J G, McElhaney J E, Verschoor C P. Reliable reference genes for the quantification of mRNA in human T-cells and PBMCs stimulated with live influenza virus[J]. BMC immunology, 2020, 21(1): 1-7.
[3] Jiang, J.L.; Qi, L.N.; Wei, Q.W.; et al. Maternal sativoside supplementation ameliorates intestinal mucosal damage and modulates gut microbiota in chicken offspring challenged with lipopolysaccharide. Food & Function 2021, 12, 6014-6028.
[4] Liu, W.C.; Huang, M.Y.; Balasubramanian, B.; et al. Heat Stress Affects Jejunal Immunity of Yellow-Feathered Broilers and Is Potentially Mediated by the Microbiome. Frontiers in Physiology 2022, 13.
[5] Gao, S.; Zhang, L.; Zhu, D.; et al. Effects of glucose oxidase and bacillus subtilis on growth performance and serum biochemical indicexs of broilers exposed to aflatoxin B1 and endotoxin. Animal Feed Science and Technology 2022, 286.
[6] Thiam, M.; Sanchez, A.L.B.; Zhang, J.; et al. Association of Heterophil/Lymphocyte Ratio with Intestinal Barrier Function and Immune Response to Salmonella enteritidis Infection in Chicken. Animals 2021, 11.
Comment 8. Statistical analysis – what kind of test did you apply for means separation?
Response: We gratefully appreciate the reviewer’s professional comments. Detailed information about the test that applies for means separation had been added in Statistical analysis part. The data separation method used in this paper is one-way ANOVA followed by Turkey’s multiple comparison tests using SPSS software (version 20.0; IBM Inc., NY, USA). Turkey's method in one-way ANOVA is suitable for the same situation in each group, so we choose this method.
Comment 9. Lines 196-203. Comparing to CTR, ANTI DID NOT significantly influenced BW at 38 day, the same was true for F/G.
Response: Many thanks for pointing out our mistake and we had revised this part.
Comment 10. Figure 1. Why the number of replicates (n=) was 5 if 6 birds was sampled? Provide exact number of n= for each analysis.
Response: We gratefully appreciate the reviewer’s professional comments. Except for the gut microbial analysis (5 samples per group), other life indicators and qPCR used 6 samples per group.
Comment 11. Lines 322-324. Move this comment or similar given previously to the discussion section or prepare mixed section.
Response: Done as requested. Many thanks for the reviewer’s kind advice.
Comment 12. General remark to the conclusion;
It has been drawn not strictly to the results obtained (i) dietary treatment improved/affected significantly only selected indices like BW and over a selected period of birds growth, (ii) you did not measure herein inflammatory response to stress factors and (iii) you investigate only chicken response thus you can not extrapolate for other species.
Response: Many thanks for the professional comments. The conclusion had been revised.

Reviewer 2 Report
The topic investigated is within the overall scope of the special issue in Antioxidants and the theme if the manuscript is of interest.
The trial was properly designed and a good amount of data have been reported.
However, the paper needs some revisions in order to further improve its quality.
I am reporting some suggestions:
- The Abstract needs to be significantly reduced;
- The Introduction section can be further improved by including additional references of recently published papers; this is to further support the Authors' statements;
- Pay attention to the acronyms used, thy must be spelled at first use (including those in Tables and Figures);
- In Table 1, please use the same decimal points for ingredients and list the feeds according to their inclusion level;
- The Conclusions need to be revised and improved.
Author Response
The topic investigated is within the overall scope of the special issue in Antioxidants and the theme if the manuscript is of interest.
The trial was properly designed and a good amount of data have been reported.
However, the paper needs some revisions in order to further improve its quality.
I am reporting some suggestions:
Response: We deeply appreciate the reviewers’ time and efforts in providing the professional comments and valuable suggestions.
Comment 1: - The Abstract needs to be significantly reduced.
Response: Done as requested. Many thanks for the reviewer’s valuable suggestions.
Comment 2: The Introduction section can be further improved by including additional references of recently published papers; this is to further support the Authors' statements;
Response: Done as requested. Many thanks for the reviewer’s valuable suggestions.
Comment 3: Pay attention to the acronyms used, thy must be spelled at first use (including those in Tables and Figures);
Response: Done as requested. Many thanks for the reviewer’s valuable suggestions.
Comment 4: In Table 1, please use the same decimal points for ingredients and list the feeds according to their inclusion level;
Response: Done as requested. Many thanks for the reviewer’s valuable suggestions.
Commen5: The Conclusions need to be revised and improved.
Response: Many thanks for the professional comments. The conclusion had been revised.

Reviewer 3 Report
1.Materials and Methods: line 125-126 The serum immunoglobulin (IgG, IgM and IgA) concentrations were determined by ELISA kits (Nanjing Jiancheng, Bioengineering Institute, Nanjing, China). Because the IgG determined should be special ELISA kits. The kits used in this study should be proved can be used in chicken, otherwise, the results of (IgG, igM and IgA) is not credible.
2.results
Table 3 the p value is 0.041 for F/G of 1-63d,however,no superscript letters was used.
Line 223 Figure 1. MMT improved cecal microbiota diversity in broilers based on OTU level. The title of Figure 1 used the result is not suitable.
Line b 261-263: broilers in the MMT and ANTI group had significantly lower DAO and D-LA levels than that in CTR group. What is DAO and D-LA. No analyzed methods were described in Materials and Methods.
Line 292 (B) Content of MPO and NO in serum was not described in Materials and Methods also.
This manuscript studied the antioxidant status by T-SOD, T-AOC, CAT, and MDA in serum, and drew the conclusion of “improving antioxidative capacity mediated by keap1-Nrf2 pathway”. It is not enough evidence for drew the conclusion of improving the antioxidant status. It should based on more antioxidant items, eg, hepatic or intestinal mucosa antioxidant status.
Author Response
Comments1: Materials and Methods: line 125-126 The serum immunoglobulin (IgG, IgM and IgA) concentrations were determined by ELISA kits (Nanjing Jiancheng, Bioengineering Institute, Nanjing, China). Because the IgG determined should be special ELISA kits. The kits used in this study should be proved can be used in chicken, otherwise, the results of (IgG, igM and IgA) is not credible.
Response: Sorry for the missing information and the detailed information had been added. Many thanks for the reviewer’s professional comments.
Comments 2: Table 3 the p value is 0.041 for F/G of 1-63d,however,no superscript letters was used.
Response: Thank you so much for pointing out the mistakes. We had revised this issue.
Comments 3: Line 223 Figure 1. MMT improved cecal microbiota diversity in broilers based on OTU level. The title of Figure 1 used the result is not suitable.
Response: Many thanks for the reviewer’s professional comments. We had revised the title of Figure 1. “Effect of MMT on gut microbiota diversities of broilers.”
Comments 4: Line b 261-263: broilers in the MMT and ANTI group had significantly lower DAO and D-LA levels than that in CTR group. What is DAO and D-LA. No analyzed methods were described in Materials and Methods.
Response: Thank you so much for pointing out the mistakes. We had revised this issue in MATERIALS AND METHODS part. “The intestinal permeability biomarkers in serum, including D-lactate acid (D-LA) and diamine oxidase (DAO), were measured a SpectraMax M5 (Molecular Devices, USA) using assay kits (Nanjing Jiancheng, Bioengineering Institute, Nanjing, China) according to the manufacturer’s instructions.”
Comments 5: Line 292 (B) Content of MPO and NO in serum was not described in Materials and Methods also.
Response: Thank you so much for pointing out the mistakes. We had revised this issue in MATERIALS AND METHODS part. “The activities of myeloperoxidase (MPO) and content of NO (nitrogen monoxide) in serum were determined using commercially available assay kit (Nanjing Jiancheng, Bioengineering Institute, Nanjing, China) according to the manufacturer’s instructions.”
Comments 6: This manuscript studied the antioxidant status by T-SOD, T-AOC, CAT, and MDA in serum, and drew the conclusion of “improving antioxidative capacity mediated by keap1-Nrf2 pathway”. It is not enough evidence for drew the conclusion of improving the antioxidant status. It should based on more antioxidant items, eg, hepatic or intestinal mucosa antioxidant status.
Response: We gratefully appreciate the reviewer’s professional comments. In fact, we had investigated the jejunum antioxidant parameters of broilers, but we didn’t put it in the submitted manuscript. The hepatic antioxidant status was not investigated in the present study. We had added the jejunum antioxidant parameters in the revised manuscript. And we had revised the conclusion.

Round 2
Reviewer 1 Report
I have no further remarks.
Reviewer 2 Report
The Authors revised their paper according to all suggestions.
Reviewer 3 Report
There is no other question.